# Lipid Profile Modulates Cardiometabolic Risk Biomarkers Including Hypertension in People with Type-2 Diabetes: A Focus on Unbalanced Ratio of Plasma Polyunsaturated/Saturated Fatty Acids

**DOI:** 10.3390/molecules25184315

**Published:** 2020-09-20

**Authors:** Ines Gouaref, Asma Bouazza, Samir Ait Abderrhmane, Elhadj-Ahmed Koceir

**Affiliations:** 1Bioenergetics and Intermediary Metabolism Team, Laboratory of Biology and Organism Physiology, Biological Sciences Faculty, University of Sciences and Technology Houari Boumediene (USTHB), BP 32, El Alia, Bab Ezzouar, 16123 Algiers, Algeria; igouaref@gmail.com (I.G.); bouazza.asma@gmail.com (A.B.); 2Diabetology Unit, University Hospital Center, Mohamed Seghir Nekkache, 244 (16208-Kouba) Algiers, Algeria; saitabderrahmane@yahoo.fr

**Keywords:** polyunsaturated/saturated fatty acid ratio, cardiometabolic risk, insulin resistance, atherogenic biomarkers, pro-inflammatory cytokines, type 2 diabetes mellitus, essential hypertension, dietary lipids, physical activity

## Abstract

Type 2 diabetes mellitus (T2DM) is associated with lipid metabolism disorder, particularly elevated plasma levels of non-esterified free fatty acids (NEFFA) and an increased cardiovascular disease risk, such as essential hypertension (H). The plasma unbalance of saturated fatty acid (SFA)/polyunsaturated fatty acid (PUFA) ratio is a likely contributor, but the mechanisms involved are not clearly elucidated. The aim of this study is to explore the association between plasma SFA/PUFA ratio and the clusters of cardiometabolic syndrome (CMS), including the atherogenic biomarkers, inflammatory status, feeding patterns, and physical activity in people with T2DM with or without essential hypertension. The study was conducted on 784 adult male and female participants, aged between 30 and 50 years, and divided into 3 groups: 100 T2DM without hypertension (D); 368 T2DM with hypertension (DM); and 316 hypertensive participants without T2DM (H). All Participants were phenotyped regarding CMS clusters according to the NCEP/ATPIII criteria. Insulin resistance was assessed by Homeostasis model assessment (HOMA model). Metabolic, atherogenic, and inflammatory parameters were analyzed by biochemical methods; NEFFA by microfluorimetry; SFA, PUFA-n6 and PUFA-n3 by gas phase chromatography. Dietary lipids and physical activity were analyzed through the use of validated questionnaires. The clusters of CMS were found in all groups. Dyslipidemia was correlated with accretion NEFFA levels in all groups, but more accentuated in the DH group (r = +0.77; *p* < 0.001). Similarly, plasma PUFA/SFA ratio and PUFA-3 level was lower, concomitantly with a higher plasma ApoB_100_/ApoA_1_ (*p* < 0.001), lipoprotein (a), homocysteine (*p* < 0.001), and pro-inflammatory cytokines (TNFα, IL-6, IL1-β) in the DH group. Likewise, the depletion of PUFA-n3/PUFA-n6 ratio is associated with the decrease of omega 3-DHA (docosahexaenoic acid) and omega 3-EPA (eicosapentaenoic acid) (*p* < 0.001). It appears that the PUFAs-n3 ratio modulates cardiometabolic risk, inflammatory state and atherogenic biomarkers. The plasma unbalanced ratio of SFA/PUFA reflects dietary fatty acids intake. The contribution of dietary lipids is undisputed. Nutritional recommendations are required to determine the fatty acids ratio (saturated and unsaturated) provided in the diet.

## 1. Introduction

In 2020, global statistics estimate the number of cases of type 2 diabetes mellitus (T2DM) will reach 425 million. This epidemiological data should increase to 700 million by 2045, mainly due to lifestyle changes, especially those involving dietary lipids [1]. People affected by T2DM are characterized by insulin resistance, obesity, being overweight, glucose intolerance, dyslipidemia, and blood pressure disorder. The interaction between hypertension and dyslipidemia in T2DM increases cardiometabolic risk factors and enhances the prevalence of ischemic cardiomyopathy [2]. Several studies show that the increase in free fatty acid plasma levels, particularly saturated fatty acids and the decrease in unsaturated fatty acids, play an important role in the development of insulin resistance by inhibiting the oxidation of carbohydrates, which accentuates T2DM [3]. The imbalance between saturated fatty acids (SFA) and polyunsaturated fatty acids (PUFA) in dietary lipids composition has been positively correlated with several metabolic disorders, characterized by inefficient insulin function in non-glucose dependent tissues (skeletal muscle, liver, and adipose tissue) and other abnormalities, such as chronic inflammation, pancreatic -cells loss and atherosclerosis [4]. Among PUFA, alpha linolenic acid (C18: 3 n3)/linoleic acid (C18: 2 n6) ratio and docosahexaenoic (DHA) omega 3 (C22: 6 n3)/eicosapentaenoic (EPA) omega 3 (C20: 5 n3) ratio [5] have been mostly studied for their beneficial vascular effects against atherosclerosis development in T2DM [6]. EPA and DHA result in hypotriglyceridemic effects by inhibiting hepatic synthesis of VLDL lipoproteins, increase HDL-c (DHA effect), and lower cholesterolemia via increased synthesis of apolipoproteinemia A1 (EPA effect) [7].

Besides, PUFA-3 modulates hypertension by incorporating into the membranes of red blood cells [8]. In contrast, the excessive SFA in dietary lipids are recognized as predictive of coronary insufficiency [9] and venous thromboses apparition [10], mainly lauric (C12: 0), myristic (C14: 0), and palmitic (C16: 0) acids. In addition, several studies have shown that atherothrombotic complications caused by lipids lead to endothelial dysfunction. These events maintain other disorders related to the plasma homocysteine and lipopoprotein (Lp (a)) levels. Indeed, hyperhomocysteinaemia and dyslipidemia are associated with increased damage to inner arteries and promote thrombosis through a pathological collagen activation pathway through increased oxidation of low-density lipoproteins [11].

Concerning Lp (a), the literature reports few studies about the relationship between Lp (a) and dietary lipids. Nevertheless, it has been described that an increase of Lp (a) is associated with a reduction of unsaturated fatty acids and an increase of saturated fatty acids [12]. In this investigation, we studied the associations between PUFA/SFA-PUFA-n3/PUFA-n6 ratios and the cardiometabolic risk factors including insulin resistance (HOMA-IR) and atherogenic biomarker status in T2DM participants with or without essential hypertension. In this study, we also investigated the associations between plasma lipid disorder and dietary lipids and physical activity.

## 2. Results

### 2.1. Cardiometabolic Risk Profile

The majority of D, H, and DH participants exhibit an obesity state. The numbers of female participants were higher in all groups compared to males (sex ratio male/female = 0.94). However, in both sexes, the waist circumference (WC) shows that adipose tissue most prominent in abdominal area (Table 1). A positive correlation exists between body mass index (BMI) and gender in all groups (r = +0.77, *p* < 0.001). The body fat percentage (% BF) is positively correlated with BMI in the DH group (r = +0.88, *p* < 0.001).

The data mentioned in Table 2 shows that the glycemia and the HbA1C (>7%) are higher by 57% in DH group versus D group (*p* < 0.001). The correlation is positive between glycemia and HbA1C in the DH groups (r = + 0.75, *p* < 0.001). Insulin resistance (HOMA-IR) was found in all participants (Table 2), but more pronounced in DH versus control group (*p* < 0.001). A strong positive correlation was found between the Homa-IR and the body fat percentage in DH and D groups versus control group (r = + 0.96, *p* < 0.001 and r = + 0.39, *p* < 0.01, respectively). On the other hand, we have noted a higher fasting circulation of insulin levels in all DH and D groups versus control (>36% and >25% respectively, *p* < 0.001).

The data summarized in Table 2 showed dyslipidemia in the DH and D groups, characterized by hypertriglyceridemia (>57% and >70%, respectively; *p* < 0.001) vs. control group. HDL-c concentration is lower (>26%, *p* < 0.05) in DH male group. Also, there is a strong positive correlation between hypertriglyceridemia and HOMA-IR test, and the plasma NEFFA level in the DH group (r = +0.69; *p* < 0.01; r = +0.77; *p* < 0.001, respectively). In addition, the LDL-c lipoprotein fraction showed a disturbance in the DH and D groups (>34% and >29%, respectively) (Table 2). The blood pressure data recorded in Table 2 is on average >24% higher in the DH group compared to the control group.

According to criteria of the World Health Organization [13], the DH and H groups can be classified as grade I, moderate hypertension (between 140/90 and 159/99 mmHg; systolic blood pressure (PAS)/diastolic blood pressure (DBP), respectively). In contrast, group D remained normal blood pressure values. The data reported in Table 2 indicates that Hs-CRP is higher in the DH group compared to D, H, and the control groups (*p* < 0.001). We have noted a positive correlation is between Hs-CRP and BF percentage (r = + 0.67, *p* < 0.01); and between Hs-CRP and HOMA-IR (r = + 0.58, *p* < 0.01). However, no fibrinogen related disorder was noted. Similarly, we observed a positive correlation between Hs-CRP and archidonic acid in the DH group (r = + 0.84, *p* < 0.001). The HDL-c/LDL-c ratio is reduced in D, H, and DH groups versus control group (Table 3). On the other hand, the TG/HDL-c ratio, the plasma levels of Lp (a), and Hcy are higher in the D and H groups, but more accentuated in the DH group vs. control group (*p* < 0.001). A positive correlation was found between Lp (a) and Hs-CRP, and between Hcy and Hs-CRP (r = + 0.79; r = + 0.85, respectively) in the DH group (Table 3).

Besides, the significant reduction of HDL-c/LDL-c ratio is associated with a raise in pro-inflammatory cytokines, particularly TNF-α, IL-6, and IL-1 (Table 3) in D and H groups, but much more in group DH. In addition, the inflammatory state is associated with the PUFA/SFA imbalance, illustrated by strongly increased SFA in group DH compared to D and H groups.

On the other hand, no association was found between SBP both with Lp (a) and Hyc in DH and D groups. In addition, we observed a negative correlation between Lp (a) and PUFA/SFA ratio (r = −0.59, *p* < 0.01) in DH group (Table 4). The correlation is positive between Lp (a) and PUFA-n6/PUFA-n3 ratio (r = +0.68, *p* < 0.01) in DH group (Table 4). On the other hand, no association was found between the PUFA/SFA ratio or PUFA-n6/PUFA-n3 ratio and Hcy in both DH and D groups. In addition, the concentrations of microalbuminuria confirm a renal disorder (microalbuminuria >30 mg/24 h) in DH group (Table 3).

### 2.2. Plasma Fatty Acids Profile

The DH participants group exhibit strongly higher plasma NEFFA levels compared to D, H, and control groups (Figure 1A, *p* < 0.001). Concomitantly, the SFA levels are excessively high in DH group versus D, H, and control groups (Figure 1B, *p* < 0.001). Among the NEFFA profile, we noted a higher content of myristic acid in DH group (Figure 2A, *p* < 0.001), palmitic acid (Figure 2B, *p* < 0.001), stearic acid (Figure 2C, *p* < 0.01) and lauric acid (Figure 2D, *p* < 0.01). The fraction of MUFA, represented in our study by oleic acid (C18: 1) is significantly reduced in DH group versus D, H, and control groups (Figure 1C, *p* < 0.001); while it remains normal in D group. Concerning plasma total PUFA, we did not observe any difference between D and H groups versus control group. On the other hand, the difference is significant in DH group versus D, H, and control groups (Figure 1D, *p* < 0.01). Besides, linoleic acid (PUFA-n6) is moderately lower in DH group versus D, H, and control groups (Figure 3A, *p* < 0.001). Similarly, linolenic acid (PUFA-n3) is heavily depleted in the DH group versus D, H, and control groups (Figure 3B, *p* < 0.001). This result is associated with the fall of the PUFA-n6/PUFA-n3 ratio (Figure 4B, *p* < 0.001).

In contrast, the arachidonic acid (PUFA-n6) is increased in D and H groups, but more markedly in group DH (Figure 3C, *p* < 0.001). The plasma EPA and DHA levels are depleted in DH group versus D, H, and control groups (Figure 4C, 4D, respectively). Furthermore, we noticed that the PUFA/SFA ratio was significantly lowered in DH group versus D, H, and control groups (Figure 4A). Similarly, the PUFA-n3/PUFA-n6 ratio is highly depleted in the DH group versus D, H, and control groups (Figure 4B, *p* < 0.001). In this context, we found a positive correlation between the drop of PUFA/SFA ratio and the lower HDL-c/LDL-c ratio (r = +66, *p* < 0.01) in DH group (Table 4). The correlation was also associated with a higher ApoB100/ApoA1 (Table 4), TG/HDL-c (r = −71, *p* < 0.001; r = −68, *p* < 0.001, respectively) and the Lp (a) levels in DH group (r = −0.5, *p* < 0.01).

### 2.3. Feeding Patterns of the Participants

#### 2.3.1. Dietary Saturated Fatty Acids (SFA) Intake

The food frequency revealed the SFA (palmitic and stearic acids) are excessively consumed, particularly in DH groups. We found 42.5 ± 1.5 g/24 h versus 10.5 ± 1.4 g/24 h in controls groups (*p* < 0.001).

#### 2.3.2. Dietary Monounsaturated Fatty Acids (MUFA) Intake

The consumption of MUFA (essentially, oleic acid in olive oil) is not significant between D, H, and DH groups versus controls groups. We have estimated 42.7 ± 2; 41.7 ± 3 and 38.8 ± 6 g/day versus 44.1 ± 9 g/day, respectively.

#### 2.3.3. Dietary Polyunsaturated Fatty Acids (PUFA) Intake

In this section, we have evaluated the consumption of Linoleic (omega 6), Linolenic (omega 3), and Arachidonic acids (omega 6), DHA and EPA (omega 3). The PUFA-linoleic acid (in sunflower oil) is most consumed, whether in DH or control groups (14 versus 7 g/day). At opposite, the PUFA-linolenic acid (in nut oil) is the fraction least consumed in DH compared to control groups (0.39 ± 0.08 versus 1.35 ± 0.05 g/day; respectively, *p* < 0.001). Interestingly, we have noted that PUFA/SFA ratio is < 2 in DH groups (1.19 ± 0.01). At opposite, this ratio is 2 in control group (3.06 ± 0.03). The PUFA-Arachidonic acid (in meats and poultry) is largely consumed in DH group comparatively with control group (475 mg/day versus 123 mg/day). Concerning the PUFA-DHA (in oily blue fish), we recorded 259 mg/day in DH group versus 430 mg/day in control group (*p* < 0.01); while in the PUFA-EPA (in oily blue fish), we observed 60 mg/day in DH group versus 120 mg/day in control group (*p* < 0.001). The results showed dietary intake ratio of n-3/n-6 PUFAs is much lower in DH group versus control group (2.28 ± 0.21 versus 5.91 ± 0.33, respectively, *p* < 0.001).

### 2.4. Physical Activity Levels of the Participants

The groups D and H participants show an insufficient physical activity between 11–22 min/day; while the DH group is mostly a sedentary lifestyle with no physical activity. This is linked to cardiac disorders which can lead to stroke during physical exercise.

## 3. Patients and Methods

### 3.1. Participant’s Inclusion and Protocol Design

This study is transversal and observational; Case-control was carried between September 2018 and October 2019. The participant’s cohort was classified according to age and sex, with a sex ratio of men/women = 0.94. This study was undertaken on type-2 diabetes participants without hypertension (D group) or with hypertension (DH group) and hypertensive participants without type-2 diabetes (H), aged between 30 and 50 years. All participants were admitted to the diabetology unit of Algiers University Hospital Center, Mohamed Seghir Nekkache (ex. Hôpital Central de l’Armée, Ain naadja). We also included in this study voluntary subjects (healthy control groups), without pathologies and non-smokers. Diabetic participants were treated with metformin 400 ± 32 mg/24 h, associated with a sulfonylurea. The DH group was treated with a variable combination therapy: beta-blocker, calcium channel blocker, inhibitor of the angiotensin converting enzyme, and diuretic. No participants were insulin-requiring. The drug doses were stable throughout the study. The age of diabetes and the presence of hypertension in DH group were variable, between 5 and 10 years. In this study, we excluded all subjects with endocrinopathies, such as Cushing’s disease, dysthyroidism, acromegaly, pheochromocytoma, pituitary, and adrenal insufficiency. We also excluded pregnant women and those on oral contraceptives. Similarly, patients treated with corticosteroids, antidepressants, hormone therapy, and type 1 diabetics were excluded. The study protocol (Algiers Diabetes Study) was approved by the Ethics Committee of the Algerian Ministry of Public Health according to the 1975 Helsinki Declaration, revised in 2008 (http://www.wma.net). An informed consent form was signed by each participant. All participants were recruited in the unit for exploring metabolic and cardiovascular diseases. All participants have been examined by the same physician including clinical investigations. During the consultation, we explored: food intake, physical activity level, other pathologies screening, anthropometric status, microangiopathies and evaluation of macrovascular complications (ultrasonography, scintigraphy, echocardiogram, and the lower limbs echo-doppler).

### 3.2. Feeding Pattern Participants

The feeding pattern of the participants is evaluated by frequency questionnaire and 24-hour recalls established by the epidemiological survey TAHINA (Transition and Health Impact in North Africa) [14]. TAHINA study was undertaken in Maghreb region: Morocco-Algeria-Tunisia with collaboration of IRD (Institut de recherche pour le développement, Montpellier, France) 

The dietary lipids were measured from the following foods: animal fat, eggs, vegetable oils, butter and margarine in cakes and pastries, fried foods, oleaginous fruits, soft drinks, and ice cream. The food frequency consummation was adapted according to El Kinany et al., 2018 [15]. The remaining percentage of dietary lipids is represented by the other lipids classes not evaluated in this study, mainly dietary cholesterol and phospholipids.

The FFQ questionnaire was entirely self-administered and had to be returned within 2 weeks after it was received by mail. The FFQ included questions on frequency and amount of intake in household measures of 139 questions on 124 food items. The questionnaires were checked for completeness after receipt and incomplete questions were supplemented by telephone interview. Besides, the dietary questionnaires were performed at the participant’s home by nutritionists. The conversion from foodstuffs to energy and nutrient intakes was established according CIQUAL composition food database [16]. Our study was based mainly on the FFQ test. The 24-h recalls method was used to approve (or not) the data obtained by the FFQ.

### 3.3. Evaluation of Physical Activity Levels in Participants

The physical activity level was assessed using the self-administered International Physical Activity Questionnaire (IPAQ) [17]. The participants were classified into four categories: (i) no regular physical activity with a sedentary lifestyle; (ii) minimal physical activity (<75 min/week); (iii) insufficient physical activity (75 min < 150 min/week); (iv) and sufficient physical activity (150 min/week). In addition to physical activity itself, the questionnaire covers four areas: work-related, transportation, housework/gardening, and leisure-time activity.

### 3.4. Cadiometabolic Syndrome (CMS) Screening

The CMS was diagnosed according to the NCEP/ATPIII (National Cholesterol Education Program Third Adult Treatment Panel) [18]. The CMS was characterized by the presence of three or more abnormal components as follows: (i) abdominal obesity; (ii) high plasma triglyceride level; (iii) low plasma HDL cholesterol level; (v) high fasting plasma glucose and (iv) a blood pressure disturbance. Insulin resistance was calculated by the homeostasis model assessment insulin resistance (HOMA-IR) method: HOMA index = fasting glucose (mmol/L) × fasting insulin (mU/L)/22.5 [19]. The percentage of body fat (BF) was calculated using the formula: (1.2 × BMI) + (0.23 × age) − (10.8 × S) − 5.4 (S is the gender correction factor) [20]. The SBP and DBP were measured in the prone position of the two arms, three times and two minutes after five minutes of rest using a validated Omron 705 CP type BP monitor (Omron Healthcare Europe BV, Amsterdam, The Netherlands) [21].

### 3.5. Blood Samples and Analyses

The participants were admitted to the hospital at 7 am after 12 h of fasting before they consumed their drugs (therapeutic treatment). Blood samples were centrifuged at 3000 rpm for 10 min, and plasma was obtained. Fasting plasma samples were immediately put on ice and kept frozen at −80 °C until analyses were performed. Fasting plasma glucose, triglycerides (TG), total cholesterol (TC), and high-density lipoprotein cholesterol (HDL-C) were determined by enzymatic methods using an automatic biochemical analyzer (Cobas Integra 400^®^ analyzer, Roche Diagnostics, Meylan, France). The low-density lipoprotein cholesterol (LDL-C) was calculated using Friedewald’s formula [LDL-C (mg/dL) = TC − HDL-C −TG/5.0] applied to subjects with CMS [22]. The high-sensitive C-reactive protein (Hs-CRP) level was assessed using immunoturbidimetric methods on chemical Synchron analyzer LX^®^20 PRO. Plasma TNF-α (Tumor necrosis factor alpha), IL-6 (Interleukin-6) and IL-1 (Interleukin-1beta) were determined using commercial enzyme-linked immunosorbent assays (ELISA) according to the manufacturers’ instructions (Cayman Chemical’s ACETM EIA kit, Ann Arbor, MI, USA, Cat N° 589,201, Cat N° 58,331, and Cat N° 583,361, respectively). The fibrinogen was evaluated by the chronometric Von Clauss methods using hemostasis analyzer ACL TOPTM (Biolabo, Maizy, France). Insulin concentrations were determined by RIA (RadioImmunoAssay) using commercially available kits (Human insulin specific RIA kit, EMD Millipore Corporation St. Louis, MO 63,103, USA). Apolipoprotein A1 (Apo A1), Apolipoprotein B100 (Apo B100) and Lp (a) lipoprotein were determined by Synchron LX^®^20 PRO analyzer. Homocysteinemia (Hcy) was assessed using FPIA (fluorescence polarization immuno assay) on Immulite 2000 analyzer Ref: L2KH02. HbA1c and Microalbuminuria were determined by immunoturbidimetry on a Cobas Integra analyzer.

### 3.6. Plasma Fatty Acids Determination

The blood samples were taken on sodium oxalate. The total plasma lipids were separated by the Folch [23] and Dole [24] methods from 0.5 mL of plasma by adding 5 mL of a chloroform/methanol mixture (2:1 *v*/*v*) and 1 mL of 5% butyl-hydroxy-toluene in methanol.

The homogenate was purified on degreased filter paper (Durieux brand without ash N° 114–110 m/m). After the extraction in the heptane phase, the dry residue containing the fatty acids was taken up in 50 μL of hexane, 1 μL of the solution obtained was injected into a stationary phase capillary column of polyethylene glycol (HP-Innowax type), 30 m length, 0.32 mm inside diameter, and 0.5 µm film thickness.

The assessment of saturated fatty acids (SFA), Monounsaturated fatty acids (MUFA), polyunsaturated fatty acids (PUFA), Eicosapentaenoic acid (EPA), and Docosahexaenoic acid (DHA) were analyzed by gas chromatography on HP5890A (Hewlett-packard-normalk analyzer) series II equipped with a flame ionization detector. The carrier gas was nitrogen with a flow rate of 1 mL/min. The injector, detector, and column temperatures were 220 °C, 275 °C and 180 °C, respectively. Adding internal standard or internal controls to samples allows the quantification of fatty acids within the sample by calculations using the area of known quantity of the internal standard peak relative to the area of the peak fatty acids. Internal standards dissolved in 1 mL/mg of dry chloroform: methanol (2:1, *v*/*v*) containing butylated hydroxytoluene (BHT; 50 mg/L) as antioxidant. The loss in the total amount of fatty acids extraction by our method is estimated between 5 and 10%. The NEFFA were extracted by the Duncombe method [25] and evaluated by microfluorimetry using a KONTRON analyzer, Power Supply SFM23, Augsburg, Germany.

### 3.7. Statistical Analysis

All statistical analyses were performed with Epi-info version 5 and Statview version 5 (Abacus Concepts, Berkeley, USA). Student’s *t*-test and one-way ANOVA were used for the comparison both between the 03 groups (D, H, and DH) and with the control group, mainly to determine the severity of CVD in the DH group. Pearson’s correlation (coefficient (r)) analysis was performed to quantify associations between CMS clusters and the other study parameters, in particular the fatty acids ratio and atherogenicity biomarkers between D and DH groups versus control group. The results were considered significant at * *p* < 0.05, very significant at ** *p* < 0.01, or highly significant at *** *p* < 0.001. The data of FFQ were analyzed using the computer program SPSS version 11 (SPSS Inc., Applications Guide, Chicago, IL, USA). Statistical methods for validation of FFQ include the Student’s *t*-test, Pearson correlation, and the kappa statistic [26].

## 4. Discussion

This study highlights the association between feeding patterns, CMS, metabolic, and hemodynamic abnormalities by elucidating the complications related to type 2 diabetes mellitus (group D) and essential hypertension (DH group). Based on lifestyle, we have observed that DH group does not follow the dietary recommendations advised by the physicians; such as to apply the Mediterranean diet (MedDiet). In our study, the PUFA/SFA ratio is less than 2; while it should be greater than 2 in the MedDiet. Indeed, several studies have shown that the MedDiet is negatively correlated with biomarkers of cardiovascular risk [27]. Similarly, we found that the ratio of PUFA-omega3/PUFA-omega 6 is less than 5 in DH group versus control group. The depletion of this ratio is also associated with cardiovascular diseases, as was demonstrated in the OPTILIP study [28]. Concerning physical activity, our study confirms the association between a sedentary lifestyle and hypertension development in the diabetic subject (DH group). According to many cohort studies and meta-analyses, the physical activity practice improves the lipid profile in T2D subjects who have become hypertensive [29]. We have shown the lipid profile disorder is associated with insulin resistance (HOMA-IR) and vascular dysfunction related to the unbalance polyunsaturated/saturated fatty acids ratio. It should be noted that abdominal adiposity with accretion of visceral adipose tissue (VAT) inaugurates all the metabolic and vascular disorders observed in D and H groups, but more marked in DH group. However, the serum NEFFA flux is found excessively high particularly in the DH group. In fact, the VAT is the origin of the significant flow of serum NEFFA, which explains the hypertriglyceridemia in the DH group. The NEFFA circulating in the D and DH groups is due to a lipolytic hyperactivity of the VAT (rich in β-adrenergic receptors) leading to a state of insulin resistance, which aggravates hypertension [30]. The NEFFA production flow maintains an insulin resistance state by altering insulin signaling [31], partly explaining the association between hyperglycemia and hypertension in diabetic participants (DH group) [32]. Several studies prove that the obesity-hypertension relationship is due to the hyperactivity of adipocyte triglyceride lipase and the inhibition of NEFFA oxidation [33]. Indeed, the Framingham study highlighted the severity of obesity as a major cardiovascular risk factor in the genesis of hypertension [34]. In addition, the hypertrophy of VAT observed in DM participants contributes to the blood pressure disorder, since it is admitted that adipose tissue is an important source of production of Angiotensin II as a powerful vasoconstriction factor [35].

In addition, the presence of microalbuminuria in the DH group is linked to the NEFFA transport by albumin. Impaired renal function results in decreased ultrafiltration of glomerular albumin. The albumin is saturated following an excess of circulating NEFFA, which leads to proteinuria [36]. According to the National Kidney Foundation [37] microalbuminuria aggravates the kidney interstitial damage in hypertension with or without type-2 diabetes [38]. In addition, lipotoxic NEFFA and urine albumin leakage seem to cause systemic inflammation (exacerbated Hs-CRP in the DH group), responsible for overproduction of pro-inflammatory cytokines, especially in the kidneys [39].

Insulin resistance (HOMA-IR) in the DH group explains the blood pressure disorder by increasing the SBP. Insulin is involved in vasoconstrictive signaling pathways, including the extracellular signal-regulated kinase pathways, mitogen activated phosphate kinase, and endothelin-1 [40]. The acute insulin resistance observed in the DH group also explains the increase in hyperglycemia which affects HbA1C (>7%) via the proteins glycation, including hemoglobin [41]. In this study, the interactions between lipid disorder and hypertension (DH group) appear to be linked to excessive formation of advanced glycation products or AGEs (Advanced Glycation End products) under the chronic effects of insulin resistance [42,43]. In DH group, the AGEs are found in the aortic valve stenosis and the peripheral arteries (data not shown).

Regarding the high levels of Lp (a) in the DH group compared to the H and D group, is explained by the renal disorder (high microalbuminuria in the DH group versus D and H groups), because it is eliminated mainly by glomerular filtration [44]. Previous data from our laboratory have shown that patients in the DH group may present with nephrotic/cardiorenal syndrome or coronary insufficiency [45]. In this context, an association has been described between Lp (a) levels and hypertension via renal endothelial dysfunction. In fact, the oxidation of Lp (a) leads to the development of atherosclerotic plaques [46]. Oxidized Lp (a) generates other oxidized metabolites via oxidative stress which interact with Toll-like receptors involved in vascular dysfunction [47]. Previous work has shown that Lp (a) can be sequestered in the arterial subintimal space much more than oxidized LDL-c, and therefore lead to atherosclerosis [48]. Lp (a) has been shown to promote vascular dysfunction in a diabetic subject by decreasing the concentrations of NO (nitric oxide), recognized as an endothelial relaxant [49]. On the other hand, the resurgence of Lp (a) in DH subjects can also be linked to drug treatment where Lp (a) is very slightly reduced by statins [50].

Lipids disorders are linked to disturbances in the fatty acid profile. The latter is involved in hemodynamic and thrombotic abnormalities by three major effects: (i) endothelial dysfunction (vaso-relaxation); (ii) atherogenic; and (iii) inflammatory state which alters blood flow. Previous studies have shown that SFA consumed in excess in food and Trans fatty acids synthesized from unsaturated fatty acids has been associated with high cardiovascular mortality following ischemic heart disease [51,52]. Several mechanisms have been proposed to explain the interactions between SFA and hypertriglyceridemia and high plasma LDL. The main incriminated are palmitic (C 16:0) and myristic (C14: 0) acids, because they increase LDL-c [53], decrease the activity of LDL receptors by reducing of Clathrin mRNA expression, and modulate ApoB [54].

In our study, palmitate represents 25% of SFA in total NEFFA, which explains the strong increase in hypertriglyceridemia observed in the DH group. In addition, plasma SFA maintains a state of insulin resistance (enhance HOMA-IR) by promoting the accumulation of lipid derivatives, such as the synthesis of ceramides that inhibit intracellular insulin signaling, principally in skeletal muscle [55].

On the other hand, SFAs reduce the HDL-c/LDL-c ratio via their influence by the PUFA/SFA ratio (DH group), promote an atherosclerotic state through the formation of dense and small LDL [56], and are associated with low-grade inflammation. Besides, the significant increase Hs-CRP is associated with higher plasma pro-inflammatory cytokines levels, particularly TNF-α, IL-6 and IL-1 (Table 3) in DH group versus D and H groups. In addition, the inflammatory state is associated with the PUFA/SFA imbalance, illustrated by strongly increased SFA in group DH compared to D and H groups. Several studies have shown that pro-inflammatory cytokines play an important role in the development of T2DM [57]. The pro-inflammatory cytokines exert a significant lipolytic effect and increase NEFFA flux which induces dyslipidemia and an atherogenic effect [58].

In this context, Palmitic acid has been described to activate intracellular pro-inflammatory pathways in human umbilical endothelial cells [59]. In addition, palmitic acid increases the expression of interleukin IL6 in macrophages and muscle cells [60]. It has also been shown that palmitic acid activates the expression of the nuclear necrosis factor NF-kB and TNFα in adipocyte lines [61]. However, it should be emphasized that the mechanisms due to the SFA effects are complex. It should be noted that not all SFA have an atherogenic effect. The stearic acid (C 18: 0) does not modify the expression of the LDL receptor. This is linked to the sensitivity of stearate to the activity of hepatic stearoyl-CoA desaturase which degrades stearate to monounsaturated fatty acid (biosynthesis of oleate from stearate). Stearic acid is poorly incorporated into triglycerides and does not increase LDL-c [62]. Various meta-analyses and epidemiological studies have demonstrated the benefit of PUFA, mainly PUFA-n3 (EPA and DHA), as agents for preventing heart failure [63]. The PUFA-n3 effects have been observed on dyslipidemia [64], blood pressure disturb via the NO decreased production [65], haemostasis disorder [66], and inflammation dysfunction [67].

In our research, we noticed a decrease in the PUFA/SFA ratio concomitantly with the depletion of linolenic (PUFA-n3), and an increase in arachidonic acids levels (PUFA-n6). These events are associated with a marked increase in linoleic acid (PUFA-n6) and can be explained by the decrease in long chain derivatives of EPA and DHA (PUFA-n3). Given the competition between the PUFA-n6 and PUFA-n3, this can be explained by the desaturation and elongation of the desaturases ∆-6 and ∆-5 pathways [68]. In this situation, the overload of PUFA-6 would regulate the synthetic pathway towards arachidonic acid (C20: 4 n6). In our investigation, arachidonic acid was found to be significantly increased in the DH group compared to the D and controls groups. This event can be explained by an overproduction of the inflammatory factors in atherosclerosis, mainly by products derived from the oxidation of lipids [69]. The phenomenon can predict arachidonate conversion due either to production of prostaglandins transformed to thromboxane A2 (vasoconstrictor and platelet aggregator) under the action of cyclooxygenase [70], or to the synthesis of leukotrienes under the action of 5-lipoxygenase [71]. Furthermore, previous studies have shown in diabetic subjects with or without coronary insufficiency, the arachidonic acid is incorporated more into the membrane phospholipids of blood platelets [72].

Moreover, it is crucial to emphasize that the DH group showed acute levels of Lp (a) compared to H and control groups and to a lesser extent in D group. This observation is associated with an increase in the ApoB100/ApoA1 ratio can predict the atheromatous and thromboembolic state in the DH group. It is essential to emphasize that the peak of Lp (a) is correlated with the decrease in the PUFA/SFA ratio and with the increase in the PUFA-n6/PUFA-n3 ratio. The literature reports very limited studies on the relation between Lp (a) and diet; however, it has been described that the elevation of serum levels in Lp (a) is associated with a unbalanced unsaturated fatty acids/ saturated fatty acids, but chiefly with high-carbohydrate diet [73]. Incontestably, Lp (a) appears to be a biomarker for predicting the risk of atherosclerosis in type-2 diabetes with hypertension [74].

In our investigation, the nutritional ratio of fatty acids in the food ration can be considered as atherogenic biomarkers. Indeed, several clinical interventional studies relating to PUFA-n3 supplementation show a beneficial effect on the modulation of the PUFA-n6/PUFA-n3 ratio [75]. Among its advantages: (i) reduce arterial hypertension values [76]; (ii) decrease coronary risk [77]; (iii) prevent atherosclerotic plaques; (iv) suppress AGE formation [78]; and inhibit the degenerative process linked to oxidative stress in atrial fibrillation [79]. In this context, several studies have shown a positive correlation between the increase in Lp (a) levels and the high fats trans fatty acids consumption and coronary insufficiency [80].

## 5. Conclusions

Our study supports a strong association between the unbalanced ratio of plasma polyunsaturated/saturated fatty acids and the major cardiometabolic clusters, including insulin resistance (higher HOMA-IR), dyslipidemia, waist circumference expansion, and blood pressure disturbance in people with T2DM with or without hypertension. A sturdy association was found between increased pro-inflammatory cytokines (TNFα, IL-6, IL1β) and elevated plasma NEFFA levels, especially saturated FFAs (mainly palmitic acid), and diminished PUFAs, essentially linolenic acid. The unbalanced ratio of SFA/PUFAs and the relevant dietary ratio of PUFA-n6/ PUFA-n3, primarily eicosapentaenoic acid (EPA)/docosahexaenoic acid (DHA) ratio, have been positively associated with atherogenic biomarkers (including increased lipoprotein (a) and hyperhomocysteinaemia). Future challenges require the in-depth investigations of lipidomic study in cellular signaling pathways and its impact on vascular tone, essentially endothelial dysfunction in people with T2DM.

## Figures and Tables

**Figure 1 molecules-25-04315-f001:**
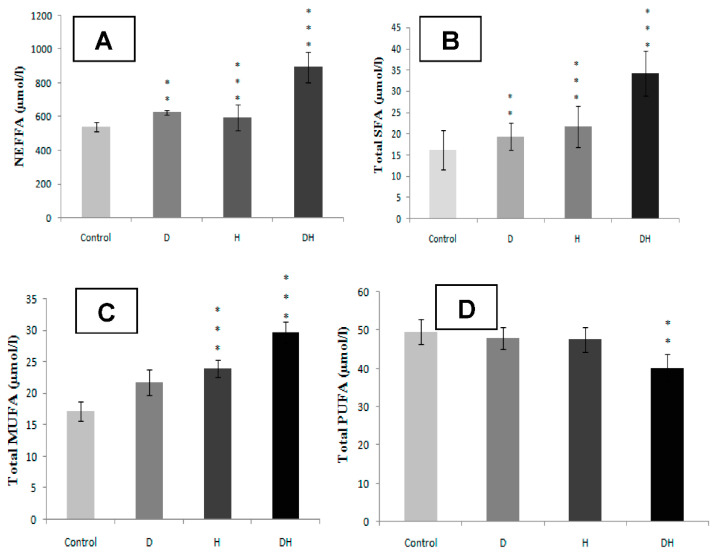
Non-esterified free fatty acids (**A**), Saturated fatty acids (**B**), Monounsaturated fatty acids (**C**), and Polyunsaturated fatty acids (**D**) plasma profile in diabetic participants groups with or without essential hypertension. ** and *** signify *p* < 0.01 and *p* < 0.001 respectively as compared DH, H and D groups to Control group.

**Figure 2 molecules-25-04315-f002:**
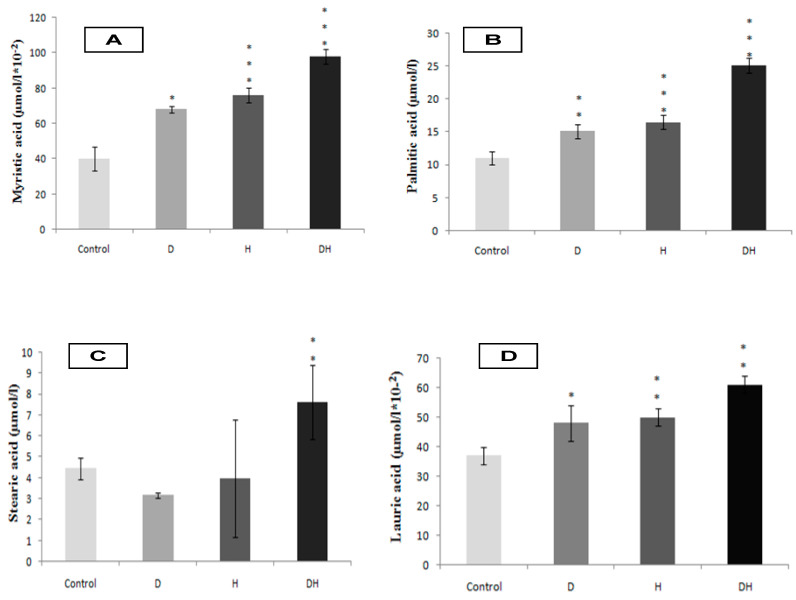
Myristic acid (**A**), Palmitic acid (**B**), Stearic acid (**C**) and Lauric acid (**D**) plasma levels in diabetic’s participants groups with or without essential hypertension. *, ** and *** signify *p* < 0.05, *p* < 0.01 and *p* < 0.001 respectively as compared DH, H, and D groups to control group.

**Figure 3 molecules-25-04315-f003:**
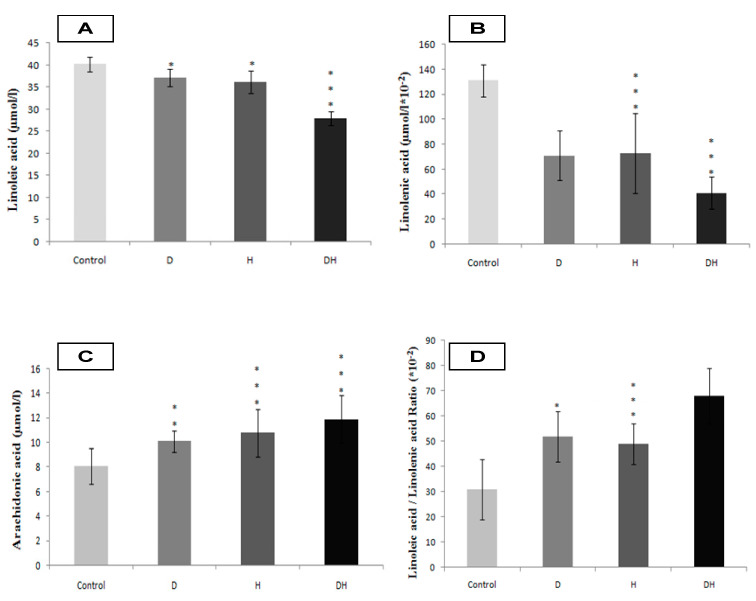
Linoleic acid (**A**), Linolenic acid (**B**), Arachidonic acid (**C**) and Linoleic acid/Linolenic acid ratio (**D**) plasma levels in diabetic’s participants groups with or without essential hypertension. *, ** and *** signify *p* < 0.05, *p* < 0.01 and *p* < 0.001 respectively as compared DH, H and D groups to Control group.

**Figure 4 molecules-25-04315-f004:**
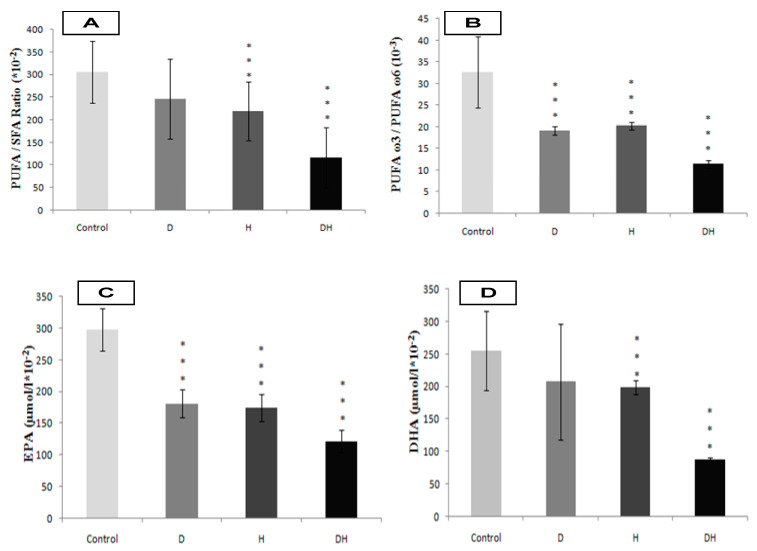
Polyunsaturated fatty acids (PUFA)/saturated fatty acids ratio (**A**), PUFA-n3 (EPA + DHA)/PUFA-n6 (Linoleic acid) ratio (**B**), Eicosapentaenoic acid (**C**), Docosahexaenoic acid (**D**) plasma levels in diabetic’s participants groups with or without essential hypertension. *** signify *p* < 0.001 as compared DH, H and D groups to Control group.

**Table 1 molecules-25-04315-t001:** Participant’s cohort distribution according anthropometric status.

*p*/G	Control	D	H	DH
	(N = 50)	(N = 100)	(N = 316)	(N = 368)
Age (year)	44 ± 3	48 ± 2	51 ± 2	45 ± 3
BMI (Kg/m^2^)	23.5 ± 2	34 ± 2 ***^(a)^	22 ± 2 ***^(a,b)^	34 ± 2 ***^(a,b,c)^
WC (cm)	77 ± 2 ^(F)^	106 ± 6 ^(F)^***^(a)^	105 ± 5 ^(F)^***^(a,b)^	111 ± 5 ^(F)^***^(a,b,c)^
	73 ± 1 ^(M)^	91 ± 3 ^(M)^	100 ± 7 ^(M)^***^(a,b)^	110 ± 7 ^(M)^***^(a,b,c)^
WC/WH ratio	0.83 ± 0.03 ^(^^F^^)^	1.09 ± 0.02 ^(F)^***^(a)^	0.97 ± 0.01 ^(F)^***^(a,b)^	1.11 ± 0.01 ^(F)^***^(a,b,c)^
	0.86 ± 0.01 ^(M)^	1.03 ± 0.01 ^(M)^	0.91 ± 0.02 ^(M)^***^(a,b)^	1.08 ± 0.02 ^(M)^***^(a,b,c)^
BF (%)	12.0 ± 0.61 ^(^^F^^)^	46.6 ± 0.2 ^(F)^***	30.51 ± 3.29 ^(F)^***^(a,b)^	52.01 ± 0.01 ^(F)^***^(a,b,c)^
	8.22 ± 0.60 ^(M)^	42.1 ± 0.9 ^(M)^***^(a)^	28.80 ± 1.10 ^(M)^***^(a,b)^	48.22 ± 0.06 ^(M)^***^(a,b,c)^

*p*: parameters; G: group; M: male; F: female; N: total number of participants; D: Type 2 diabetes mellitus (T2DM); H: Hypertension; DH: T2DM + hypertension; BMI: Body mass index; WC: Waist Circumference; WH: Waist Hips; BF: body fat percentage. The mean values are assigned from the standard error to the mean (X ± SD). The degree of significance is calculated for a risk of error α = 5%. The comparison of means is established both between the 03 groups (D, H and DH) and with the control group. *: *p* < 0.05; **: *p* < 0.01; ***: *p* < 0.001. a, b, c, d are superscript letters assigned for each group shows a significant difference as analyzed by one-way ANOVA to compare the 03 groups between them: Control (a), D (b), H (c) and DH (d). For example: DH ***^(a,b,c)^ means that the difference is highly significant (*p* < 0.001) between DH group with control, D and H groups. [normal range/risk range]: WC [<94 cm in male and <80 cm in female/>102 cm in male and >88 cm in female]; WC/WH ratio [<1 in male and female/>1 in male and female]; BF [<15% in male and <20% female / >15% in male and >20% female].

**Table 2 molecules-25-04315-t002:** Metabolic, cardiovascular, and inflammatory characterization in diabetic participants groups with or without essential hypertension.

*p*/G	Control	D	H	DH
Glycemia (mmol/L)	4.71 ± 0.12	7.55 ± 0.8 **^(a)^	5.63 ± 0.8 ***^(a,b)^	8.33 ± 0.6 ***^(a,b,c)^
Insulinemia (pmol/L)	66 ± 1.7	147 ± 2.2 **^(a)^	136 ± 3.1 ***^(a,b)^	164 ± 8
Homa Index	1.98 ± 0.07	4.19 ± 0.10 ***^(a)^	5.95 ± 0.64 ***^(a,b)^	6.97 ± 0.87 ***^(a,b,c)^
HbA1C (%)	5.1 ± 0.59	6.33 ± 0.12 **^(a)^	5.94 ± 0.10 ***^(a,b)^	9.21 ± 0.40 ***^(a,b,c)^
Triglycerides (mmol/L)	1.16 ± 0.31	1.82 ± 0.22 ***^(a)^	1.89 ± 0.45 ***^(a,b)^	1.98 ± 0.26 ***^(a,b,c)^
Total Cholesterol (mmol/L)	4.05 ± 0.1	5.27 ± 0.6 ***^(a)^	5.11 ± 0.7 ***^(a,b)^	5.61 ± 0.8 ***^(a,b,c)^
HDL-C (mmol/L)	1.52 ± 0.2 ^(^^F^^)^	1.08 ± 0.3 ^(F)^	1.10 ± 0.4 ^(^^F)^*^(a,b)^	1.04 ± 0.1 ^(^^F)^*^(a,b,c)^
	1.24 ± 0.1 ^(M)^	1.06 ± 0.2 ^(M)^	1.09 ± 0.1 ^(M)^*^(a,b)^	0.98 ± 0.1 ^(M)^*^(a,b,c)^
LDL-C (mmol/L)	2.45 ± 0.5	3.48 ± 0.6 **^(a)^	3.23 ± 0.5 ***^(a,b)^	3.69 ± 0.3 ***^(a,b,c)^
SBP (mm Hg)	121 ± 12	129 ± 6	147 ± 3 ***^(a,b)^	159 ± 3 ***^(a,b,c)^
DBP (mm Hg)	73 ± 5	80 ± 4	92 ± 1 ***^(a,b)^	98 ± 1 ***^(a,b,c)^
Hs-CRP (mg/L)	3.5 ± 1.2	5.6 ± 0.1 **^(a)^	5.4 ± 0.7 ***^(a,b)^	7.7 ± 0.6 ***^(a,b,c)^
Fibrinogen (mg/L)	2.97 ± 0.41	3.38 ± 0.13	3.19 ± 0.11	3.42 ± 0.1
Microalbuminuria (mg/24h)	14.7 ± 5	29.9 ± 14 ***^(a)^	34.1 ± 8.2 ***^(a,b)^	44.8 ± 7 ***^(a,b,c)^

HOMA: Homeostasis Model Assessment; C: cholesterol; HDL: high density lipoprotein; LDL: low density lipoprotein; SBP: systolic blood pressure; DBP: diastolic blood pressure. Hs-CRP: High sensitive C reactive Protein; M: male; F: female;. The comparison of means is established both between the 03 groups (D, H and DH) and with the control group. *: *p* < 0.05; **: *p* < 0.01; ***: *p* < 0.001. a, b, c, d are superscript letters to compare the 03 groups between them: Control (a), D (b), H (c) and DH (d). [normal range/risk range]: Glycemia [3.9–5.55 mmol/L/>6.10 mmol/L]; Insulinemia [13.8–118 pmol/L/>138 pmol/L]; HOMA-IR [0.744–2.25/>3]; HbA1_C_ [4–6%/>6%]; Triglycerides [0.68–1.48 mmol/L/>1.71 mmol/L]; Total Cholesterol [4.13–5.13 mmol/L/>5.16 mmol/L]; HDL-C [1.04–1.81 mmol/L/<1.04 mmol/L in male and <1.29 mmol/L in female]; LDL-C [1.01–3.97 mmol/L/>4.13 mmol/L]; SBP [110–120 mmHg/>130 mmHg]; DBP [70–80 mmHg/>85 mmHg]; Hs-CRP [<5 mg/L/>6 mg/L]; Fibrinogen [2–4 g/L/>5 g/L]; Microalbuminuria [<30 mg/24 h/>30 mg/24 h].

**Table 3 molecules-25-04315-t003:** Plasma atherogenic and pro-inflammatory cytokines biomarkers profile in diabetic participants groups with or without essential hypertension.

*p*/G	Control	D	H	DH
HDL-c/LDL-c	0.56 ± 0.04	0.31 ± 0.01 **^(a)^	0.33 ± 0.02 ***^(a,b)^	0.27 ± 0.02 ***^(a,b,c)^
TG/HDL-c	0.86 ± 0.01	1.69 ± 0.05 **^(a)^	1.68 ± 0.22 ***^(a,b)^	1.96 ± 0.06 ***^(a,b,c)^
ApoA1 (g/L)	1.69 ± 0.03	1.38 ± 0.07	0.87 ± 0.02 ***^(a,b)^	0.90 ± 0.05 ***^(a,b,c)^
ApoB100 (g/L)	0.85 ± 0.01	0.92 ± 0.08	0.90 ± 0.05	0.98 ± 0.08
ApoB100/ApoA1	0.50 ± 0.01	0.66 ± 0.01	1.03 ± 0.02 ***^(a,b)^	1.08 ± 0.06 ***^(a,b,c)^
Lp (a) (g/L)	0.20 ± 0.07	0.44 ± 0.06 **^(a)^	0.79 ± 0.03 ***^(a,b)^	0.83 ± 0.01 ***^(a,b,c)^
tHcy (µmol/L)	10.4 ± 0.99	10.7 ± 0.06 *^(a)^	15.2 ± 0.35 **^(a,b)^	17.1 ± 0.15 **^(a,b,c)^
TNF-α (pg/mL)	26.1 ± 2.21	55.8 ± 3.27 *^(a)^	85.5 ± 5.13 **^(a,b)^	149 ± 17.5 ***^(a,b)^
IL-6 (pg/mL)	42.6 ± 1.34	107 ± 8.5 **^(a)^	113 ± 7.61 **^(a)^	228 ± 32 ***^(a,b,c)^
IL-1β (pg/mL)	32.5 ± 4.5 **	82.1 ± 2.5 *^(a)^	109 ± 21.2 **^(a,b)^	135 ± 19.4 ***^(a,b,c)^

TG: Triglycerides; ApoB_100_: Apoprotein B; ApoA_1_: Apoprotein A; Lp (a): Lipoprotein (a); tHcy: Total homocysteine. TNF-α: Tumor necrosis factor-alpha; IL-6: Interleukine-6; IL-1β: Interleukine-1β. The comparison of means is established both between the 03 groups (D, H and DH) and with the control group. *: *p* < 0.05; **: *p* < 0.01; ***: *p* < 0.001. a, b, c, d are superscript letters to compare the 03 groups between them: Control (a), D (b), H (c) and DH (d). [normal rang/risk range]: HDL-c/LDL-c [>0.40/<0.50]; TG/HDL-c [<0.90/>1]; ApoA1 [1.10–1.80 g/L/<1 g/L]; ApoB100 [0.50–1.50 g/L/<0.50 g/L]; ApoB100/ApoA_1_ [<0.90/>0.90 or Apo A_1_/B_100_ <1.10/>1.10]; Lp (a) [<0.30 g/L/>0.30 g/L]; tHcy [5–15 µmol/l/>15 µmol/l]; TNF-α [42–103 pg/mL/>140 pg/mL]; IL-6 [30–149 pg/mL/>200 pg/mL]; IL-1β [18–107 pg/mL/>130 pg/mL].

**Table 4 molecules-25-04315-t004:** Pearson correlation between PUFA/SFA ratio and various cardiometabolic risk clusters in diabetic participants groups with or without essential hypertension.

*p*/G	Group D		Group H		Group DH	
	r	*p*	r	*p*	r	*p*
PUFA/SFA ratio—Homa-IR	−0.74	0.001	−0.85	0.001	−0.55	0.05
PUFA/SFA ratio—%BF^(F)^	0.92	0.001	−0.21	0.049	−0.59	0.05
PUFA/SFA ratio—%BF^(M)^	−0.34	0.055	0.56	0.05	−0.42	0.024
PUFA/SFA ratio—WC ^(F)^	−0.32	0.058	−0.48	0.032	−0.23	0.05
PUFA/SFA ratio—WC ^(M)^	−0.89	0.031	−0.78	0.024	−0.78	0.001
PUFA/SFA ratio—Triglycerides	−0.35	0.03	−0.2	0.055	−0.85	0.001
PUFA/SFA ratio—Total Cholesterol	−0.4	0.05	−0.6	0.01	−0.71	0.001
PUFA/SFA ratio—HDL-c/LDL-c	0.45	0.05	−0.24	0.05	0.66	0.001
PUFA/SFA ratio—TG/HDL-c	0.41	0.04	−0.12	0.005	−0.68	0.001
PUFA/SFA ratio—ApoB100/ApoA1	−0.85	0.001	−0.45	0.05	−0.71	0.001
PUFA/SFA ratio—NEFFA	−0.75	0.001	0.52	0.03	−0.2	0.02
PUFA/SFA ratio—Lp (a)	−0.57	0.02	−0.64	0.001	−0.5	0.01
PUFA/SFA ratio—tHcy	−0.42	0.05	−0.5	0.02	−0.59	0.01
PUFA/SFA ratio—Hs-CRP	−0.75	0.001	−0.23	0.051	−0.43	0.04
PUFA/SFA ratio—EPA + DHA	0.14	0.006	0.21	0.075	0.54	0.02

*p*: Parameters; G: Group; Homa-IR: homeostasis model assessment of insulin resistance. NEFFA: Non-esterified free fatty acids; SFA: saturated fatty acids; PUFA: polyunsaturated fatty acids; EPA: Eicosapentaenoic acid; DHA: Docosahexaenoic acid; M: male; F: female. The analysis correlation between two parameters was determined by Pearson’s coefficient (r). The results were considered significant (* *p* < 0.05), very significant (** *p* < 0.01) or highly significant (*** *p* < 0.001).

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
