# Peer review of "Lipid Profile Modulates Cardiometabolic Risk Biomarkers Including Hypertension in People with Type-2 Diabetes: A Focus on Unbalanced Ratio of Plasma Polyunsaturated/Saturated Fatty Acids"

_molecules, 2020, doi:10.3390/molecules25184315_

Round 1
Reviewer 1 Report
The article intituled Lipid profile modulates Cardiometabolic biomarkers including hypertension and Thrombogenic risk in Type 2 diabetic participants: Focus on Plasma Unbalance Polyunsaturated / Saturated Fatty Acids Ratio, contains interesting results about the correlation of circulating saturated and fatty acids and their correlation with diabetes and indicates the possibility of these levels being used as disease markers.
Specific comments:
The title indicates cardiometabolic and thrombotic risks, but the main evidence is about cardiometabolic risk, soI would that the thrombotic risk from the title
Abstract:
The first sentence is confusing do not describing the correlations that are done in the article. Please rephrase.
Introduction:
In the introduction, the authors could add, the importance of some markers that were used to prove their hypothesis. Some of then where in the discussion section, like HOMA-R, Lpc...
Methods:
In the plasma fatty acids detection methods, it should be added, at least in a supplemental section, a graphic of the chromatography, ou an explanation of the controls of the extraction, like internal controls, example, what is the loss in the total amount of fatty acids in their extraction method in their lab. Even though the methods are described in literature their efficiency may vary from lab to lab, and according to the sample.
The article mentions that it was observed feeding patterns of the participants, but there are no comments in the discussion about that. Since there were one blood sample and the food intake patterns are directly involved in the results it would be an interesting addition to the discussion section.
The authors used as a control group of healthy subjects, that could be very interesting to compare the results with a group of obese, but non-diabetic or hypertensive group.
Is there any analysis of the cardiovascular state of the patients, like, ECG, carotid doppler?
Results:
the results are interesting, but the authors chose a group where the disease was mild, since in D and H groups most of the levels, even though higher than control, are still in the normal range. It could be interesting to include in the tables, and not only in the references, the normal ranges and the risk ranges for each factor. Because the levels will be altered and be more or less specific as markers depending on the condition of the patient.
Table 2 fibrinogen, instead of fibrinogene
Table 3, even though the factors described may be thrombogenic, they are mainly atherogenic, they can not be considered directly thrombogenic. Please observe that no direct thrombogenic factor was determined, as Tissue Factor, for example. TNF alfa was also a good factor to determine increased thrombotic possibility, as is IL-6. CRP, as the author used is a good marker for CV risk para if observable, control group presents a high level, within the higher side of the normal range, so another parameter may be added.
The usage of data presentation as table form is easy but would be easily analyzed if demonstrated as a graph, mainly the ratio results, since they are the main results that can be used as markers as suggested.
It would be interesting to add the statistical analysis between the 3 groups, not only in relation to control. Mainly to determine the severity of CVD in the DH group.
Discussion:
The authors stated, "We have shown that lipid profile disorder is associated with Insulin resistance and vascular dysfunction via the plasma fatty acids unbalance polyunsaturated/saturated fatty acids ratio". But, there is no direct indication of vascular dysfunction, like atheroma plates, levels of NO, even though diabetes and hypertension cause vascular disfunction the author did do not show it directly.
In the same paragraph, the stated that abdominal adiposity initiates cardiometabolic diseases in DH group, but all groups presented abdominal adiposity, but table 1 did no demonstrated big differences between WC in D, H, or DH groups.
In the first paragraph the author changed the denomination DH to DM several times, please correct it. The same occurs in the second and third paragraph.
There are some terms that should not be used in the discussion like:
fulminant hypertriglyceridemia (Why fulminant?), in general, triglycerides levels, rise along time.
The most incriminated SFA (should use, the main or the principal)
In the conclusion section, even though the levels indicate that FFA ratio alterations are present in the patients, the authors do no suggest the efficiency to establish ratios as control alert to CVD when compared to the other measurements to prevent this disease.
Author Response
Point-by-point rebuttals / answers to the reviewer’s comments
[Molecules] Manuscript ID: molecules-909556 - Major Revisions
Date of submission: 07/08/2020 - Date of the reply to the submission: 28/08/2020 by the journal
Revised file within 10 days (07/09/2020)
_____________________________________________________________________
Reviewer #1
Title: Lipid profile modulates Cardiometabolic biomarkers including hypertension and Thrombogenic risk in Type 2 diabetic participants: Focus on Plasma Unbalance Polyunsaturated / Saturated Fatty Acids Ratio
Authors: Ines GOUAREF (igouaref@gmail.com), Elhadj-Ahmed KOCEIR * (e.koceir@gmail.com)
Bioenergetics and Intermediary Metabolism team, Laboratory of Biology and Organism Physiology, Biological Sciences faculty, University of Sciences and Technology Houari Boumediene (USTHB), BP 32, El Alia, Bab Ezzouar, 16123, Algiers, Algeria;
* Corresponding author
General Comments
. English language and style are fine/minor spell check required.
. Does the introduction provide sufficient background and include all relevant references?....... Yes
. Is the research design appropriate?............................... Can be improved
. Are the methods adequately described?........................ Can be improved
. Are the results clearly presented?...................................... Can be improved
. Are the conclusions supported by the results? ....................... Must be improved
Comments and Suggestions for Authors
The article intituled Lipid profile modulates Cardiometabolic biomarkers including hypertension and Thrombogenic risk in Type 2 diabetic participants: Focus on Plasma Unbalance Polyunsaturated / Saturated Fatty Acids Ratio, contains interesting results about the correlation of circulating saturated and fatty acids and their correlation with diabetes and indicates the possibility of these levels being used as disease markers.
SPECIFIC COMMENTS:
Title
- COMMENT: The title indicates cardiometabolic and thrombotic risks, but the main evidence is about cardiometabolic risk, so I would that the thrombotic risk from the title
- ANSWER: We do appreciate and fully agree with this comment.
The “thrombogenic risk” key word was chosen because in this study, we have evaluated some parameters admitted as atherogenic factors but also thrombogenic factors, such as lipoprotein (a), homocysteine, HDL / LDL, TG / HDL and ApoB100 / ApoA1 ratios. Nevertheless, in the new version of the MS we have reformulated the title by removing the word “thrombogenic risk” and maintaining only the cardiometabolic risk, as suggested by the reviewer. Please see the Track changes Manuscript (MS).
Abstract
- COMMENT: The first sentence is confusing do not describing the correlations that are done in the article. Please rephrase.
2. ANSWER: I'm sorry for missing detailed the results description in abstract. In fact, the journal's instructions authors limit to 200 words. Please see the Abstract completely revised.
Introduction
- COMMENT: In the introduction, the authors could add, the importance of some markers that were used to prove their hypothesis. Some of then where in the discussion section, like HOMA-IR, Lp (a)
- ANSWER: I agree. We sincerely appreciate this concern. We have added the importance of insulin resistance (Homa-IR), homocysteine and lipoprotein (a) related to the lipid disorder in introduction section completely rewritten. Please see the revised MS.
Methods
- COMMENT 1: In the plasma fatty acids detection methods, it should be added, at least in a supplemental section, a graphic of the chromatography, or an explanation of the controls of the extraction, like internal controls, example, what is the loss in the total amount of fatty acids in their extraction method in their lab. Even though the methods are described in literature their efficiency may vary from lab to lab, and according to the sample.
- ANSWER 1: Thank you for this helpful question that we had not considered before. In fact, the determination of plasma fatty acids was carried form another chemistry research unit specializing in chromatography, but not in our medical laboratory. In our study, the loss of fatty acids is linked to the method which is not very efficient compared to the GCSM based on deuterated isotope dilution method, which is much more precise (Quehenberger O. et al,. High Sensitivity Quantitative Lipidomics Analysis of Fatty Acids in Biological Samples by Gas Chromatography-Mass Spectrometry. BBA, 2011 1811(11): 648–656. doi:10.1016/j.bbalip.2011.07.006). The loss in the total amount of fatty acids extraction method is estimated between 5 and 10%. This argument is added in the methods section. Please see the revised MS.
- COMMENT 2: The article mentions that it was observed feeding patterns of the participants, but there are no comments in the discussion about that. Since there were one blood sample and the food intake patterns are directly involved in the results it would be an interesting addition to the discussion section.
- ANSWER 2: Thank you for the constructive comments. We added the dietary questionnaire in methods section, data in results and discussed it. Please see the revised MS.
- COMMENT 3: The authors used as a control group of healthy subjects, that could be very interesting to compare the results with a group of obese, but non-diabetic or hypertensive group.
- ANSWER 3: Sincerely, thank you for this very interesting suggestion. If the pages number is not limited by the journal (author’s instructions), I will add another obese study paragraph, since we have the results of all parameters presented in this study. If the editor-in chief allows it, we can add a new values column and discuss them. Nevertheless, the reviewer can to consult our recent papers on the obese subject (https://pubmed.ncbi.nlm.nih.gov/32681271/; https://pubmed.ncbi.nlm.nih.gov/32536220/).
- COMMENT 4: Is there any analysis of the cardiovascular state of the patients, like, ECG, carotid doppler?
- ANSWER 4: This is an important comment. The data exists in clinical records of our medical service. Unfortunately, the high number of parameters to be interpreted (our submission), we didn’t take account it. The clinical data allowed us to distinguish H and HD subjects with regard to the severity of the cardiovascular complications only for therapeutic objective, but not to research. However, if it is important to enclose the interpretations of the electrocardiograms and the carotid doppler (intima-media thickness) we can send them to the journal.
Results
- COMMENT 1: the results are interesting, but the authors chose a group where the disease was mild, since in D and H groups most of the levels, even though higher than control, are still in the normal range. It could be interesting to include in the tables, and not only in the references, the normal ranges and the risk ranges for each factor. Because the levels will be altered and be more or less specific as markers depending on the condition of the patient.
- ANSWER 1: Thank you for this constructive comment. We assessed the cardiovascular risk compared to research laboratory standards. Indeed, we have noticed to link the assay kit origin (manufactured) for each parameter analyzed. Following for your suggestion, we have included in the tables the references ranges and the risk ranges for each factor.
- COMMENT 2: Table 2 fibrinogen, instead of fibrinogene.
9. ANSWER 2: I agree. It's corrected. Please see the Table 2 in Track changes MS.
- COMMENT 3: Table 3, even though the factors described may be thrombogenic, they are mainly atherogenic, they can not be considered directly thrombogenic. Please observe that no direct thrombogenic factor was determined, as Tissue Factor, for example. TNF alfa was also a good factor to determine increased thrombotic possibility, as is IL-6. CRP, as the author used is a good marker for CV risk para if observable, control group presents a high level, within the higher side of the normal range, so another parameter may be added.
10. ANSWER 3: Thank you for this very relevant suggestion. We intend to write another article with the inflammation markers and cardiovascular risk. We add the TNF, IL6 and IL1b data. Please see the Table 3 and in Track changes MS (methods, results, discussion). Although we believe to the thrombogenic effect, as suggested by the reviewer, we maintained only the atherogenic factor and remove the thrombogenic factor. Please see the Track changes MS. - COMMENT 4: The usage of data presentation as table form is easy but would be easily analyzed if demonstrated as a graph, mainly the ratio results, since they are the main results that can be used as markers as suggested.
11. ANSWER 4: I agree. This is an important suggestion. Please see the figures added in the revised MS. Consequently, the Table 4 is removed in MS and replaced by the correlation table. - COMMENT 5: It would be interesting to add the statistical analysis between the 3 groups, not only in relation to control. Mainly to determine the severity of CVD in the DH group.
- ANSWER 5: I totally agree. We have added in the methods section (statistical analysis) and in the results tables, the superscript letters to compare the 03 groups between them: Control (a), D (b), H (c) and DH (d). Please see the table 1, 2 and 4 in Track changes MS.
Discussion
- COMMENT 1: The authors stated, "We have shown that lipid profile disorder is associated with Insulin resistance and vascular dysfunction via the plasma fatty acids unbalance polyunsaturated/saturated fatty acids ratio". But, there is no direct indication of vascular dysfunction, like atheroma plates, levels of NO, even though diabetes and hypertension cause vascular dysfunction the author did do not show it directly.
13. ANSWER 1: I agree. Thank you for this very relevant comment. As we explained previously, the data in medical imaging allow concluding with the thrombogenic effect with the lipidic disorder, particularly the fatty acids imbalance; but as we were limited by the pages number, we could not integrate the clinical observations, only metabolic results. However, since we have added the pro-inflammatory cytokines data, this is an additional argument.
- COMMENT 2: In the same paragraph, the stated that abdominal adiposity initiates cardiometabolic diseases in DH group, but all groups presented abdominal adiposity, but table 1 did no demonstrated big differences between WC in D, H, or DH groups.
14. ANSWER 2: Thank you for this very interesting comment. Although the waist circumference is higher according NCEP-ATPIII references in D, H and DH groups versus healthy group. If we observed the adiposity index (BF%), it is significantly higher in the DH group comparatively with D and H groups (Table 1), but mostly a positive correlation with higher plasma NEFFA which is exacerbated in the DH group versus D and H groups (Figure 1A). Thus, the association of metabolic disorders (diabetes effects) and vascular disturb (hypertension effects) will be more acute events in the DH group versus D and H groups. - COMMENT 3: In the first paragraph the author changed the denomination DH to DM several times, please correct it. The same occurs in the second and third paragraph.
15. ANSWER 3: I’m very sorry for this unintentional confusion. It's corrected. Please see the revised MS - COMMENT 4: There are some terms that should not be used in the discussion like: fulminant hypertriglyceridemia (Why fulminant?), in general, triglycerides levels, rise along time
- ANSWER 4: I apologize for the confusion of the term "fulminant" definition. I agree that dyslipidemia (triglycerides and cholesterol) evolves rise along time since both pathologies are chronic (diabetes and hypertension), but at the last stage of micro and macroangiopathy complications, the radiological images observed (regrettably not shown) become striking and we use the term "fulminant". As suggested by the reviewer, we remove fuminant and replaced by “strong rise” or “acute” (I hope accepted). Please see the revised MS.
- COMMENT 5: The most incriminated SFA (should use, the main or the principal)
17. ANSWER 5: I agree. Thank you very much for the meaning and precision of the scientific terms. In Algeria, we write mainly in French. It's corrected. Please see the revised MS
Conclusion
- COMMENT: In the conclusion section, even though the levels indicate that FFA ratio alterations are present in the patients, the authors do no suggest the efficiency to establish ratios as control alert to CVD when compared to the other measurements to prevent this disease.
- ANSWER: I totally agree. We have completely rewritten the conclusion where we added diet and physical activity factors in addition to the lipids disorder (fatty acids ratio). Please see the revised MS

Reviewer 2 Report
The paper by Gouaref and Koceir aims to investigate if ratio of different fats and fatty acids in the plasma modulates the cardiometabolic biomarkers in people with T2D with or without hypertension. My feedback is as per below:
Title
- Title seems to have a grammatical error. ‘Type 2 diabetic’ should be replaced with people with type-2 diabetes. And ‘focus on plasma unbalance PUFA/SFA ratio’ should be replaced with ‘A focus on unbalanced ratio of plasma PUFA/SFA’.
Abstract
- The abstract provides a very vague summary of results. Firstly, the terms increase and decrease should be replaced with higher and lower. Secondly a magnitude of change and a p-value should be stated. Was it higher by 5% or 50%? Was it significant?
- Similarly when stating correlations in the abstract please provide an r value and a p value for an accurate interpretation.
- Define Homa-IR and Lp(a) in the abstract.
Introduction
- The rationale for the study is not clear and a hypothesis is not stated.
- Please clarify the aim. Is it to look at associations or is it to predict presence of cardiovascular disease in the selected population? The different terminology used in title, aim and conclusion is confusing.
- The first sentence is too complex and grammatically incorrect. Please revise. There are several other grammatical errors in the introduction.
- In line 39 the term food lipids should be replaced with dietary lipids.
- Line 41 states various fats are correlated with a number of conditions. Correlated how? Do all have a positive correlation or a negative correlation?
Methods
- How was the number of study participants decided? Was there a power calculation done?
- How was the dietary and nutrient info collected? Using 24 hour recall or FFQ? What foods were the participants consuming? What was the dietary total fat, PUFA, MUFA and SFA intake of all participants?
- What were the physical activity levels of the participants?
- Please report the brand of the insulin kit used.
- Please replace the term diabetic patients to people with diabetes throughout.
- Line 144 'hypertension with diabetic' is grammatically incorrect and should be replaced with hypertension with or without type-2 diabetes.
Results
- Throughout the results the term increased or decreased should be replaced with higher or lower. When comparing different groups of people the term increase would be incorrect. And increase or decrease is measured from a baseline value not between groups.
- In Table 1 please give a total number of participants (N) in each group.
- Why are the participant characteristics been reported separately for males and females? The aim of the study was not to look at sex differences and not all results have been reported separately for male and females. All results should be reported combined for both genders unless the aim is to look at sex differences.
- The term Man and Women should be replaced with males and females in table 1.
- A lot of the interpretation of results is based on correlation analysis. I would suggest to make a table for all the correlations that were calculated in the analysis and provide p values in that table.
- Also a lot of comparison is done between D, H and DH groups, however the result tables only show statistics ‘compared to control group’. The authors could use a superscript letter method of showing the statistical differences between various groups?
- Line 59 and 60. It is not clear what the terms dominant adipose tissue and female predominance mean?
- % of body fatness should be replaced by body fat percentage (% BF).
- Line 65 and 91 an r value has been reported but not a p value? Please report the p values.
- Line 68 increased insulin secretion should be replaced by fasting circulating insulin levels. The authors did not measure secretion.
- Table 4 should specify ‘plasma’ in the title.
Discussion
- Again there are many errors in the use of English language in the discussion.
- I am not convinced by the conclusion. While I agree the study shows an association between the FA ratios in the plasma and some of the metabolic complications, the authors cannot prove the unbalanced ratio occurred before the metabolic syndrome symptoms. So it’s not possible to prove that the plasma FA ratios are ‘predictive’. There is just an association.
Author Response
Point-by-point rebuttals / answers to the reviewer’s comments
[Molecules] Manuscript ID: molecules-909556 - Major Revisions
Date of submission: 07/08/2020 - Date of the reply to the submission: 28/08/2020 by the journal
Revised file within 10 days (07/09/2020)
__________________________________________________________________
Reviewer #2
Title: Lipid profile modulates Cardiometabolic biomarkers including hypertension and Thrombogenic risk in Type 2 diabetic participants: Focus on Plasma Unbalance Polyunsaturated / Saturated Fatty Acids Ratio
Authors: Ines GOUAREF (igouaref@gmail.com), Elhadj-Ahmed KOCEIR * (e.koceir@gmail.com)
Bioenergetics and Intermediary Metabolism team, Laboratory of Biology and Organism Physiology, Biological Sciences faculty, University of Sciences and Technology Houari Boumediene (USTHB), BP 32, El Alia, Bab Ezzouar, 16123, Algiers, Algeria;
* Corresponding author
General Comments
. Extensive editing of English language and style required
. Does the introduction provide sufficient background and include all relevant references?.....................................................……………...Must be improved
. Is the research design appropriate?.................................... Can be improved
. Are the methods adequately described?............................... Can be improved
. Are the results clearly presented?.................................... Must be improved
. Are the conclusions supported by the results? ................. Must be improved
The paper by Gouaref and Koceir aims to investigate if ratio of different fats and fatty acids in the plasma modulates the cardiometabolic biomarkers in people with T2D with or without hypertension. My feedback is as per below:
SPECIFIC COMMENTS:
Title
- COMMENT: Title seems to have a grammatical error. ‘Type 2 diabetic’ should be replaced with people with type-2 diabetes. And ‘focus on plasma unbalance PUFA/SFA ratio’ should be replaced with ‘A focus on unbalanced ratio of plasma PUFA/SFA’
- ANSWER: We do appreciate and fully agree with this comment.
As suggested, we corrected the title. Please see the Track changes MS.
Abstract
- COMMENT 1: The abstract provides a very vague summary of results. Firstly, the terms increase and decrease should be replaced with higher and lower.
2. ANSWER 1: Thank you very much for the meaning and precision of the scientific terms. The abstract has been completely rewritten and added other data required by the reviewer1: Feeding pattern, physical activity and pro-inflammatory cytokines. Please see the Track changes MS.
- COMMENT 2: Secondly a magnitude of change and a p-value should be stated. Was it higher by 5% or 50%? Was it significant?
3. ANSWER 2: Thank you for the precision of this comment. We added the p-value to all the comparisons data between the groups. We have clarified whether the difference is significant or not. Please see the Track changes MS.
- COMMENT 3: Similarly when stating correlations in the abstract please provide an r value and a p value for an accurate interpretation.
- ANSWER 3: Thank you again for this comment. We have added the value of p-value to all the correlations ("r" value) in revised abstract and in revised MS. Please see the Track changes MS.
- COMMENT 4: Define Homa-IR and Lp(a) in the abstract
- ANSWER 4: We apologize for not defined the terms: Homa-IR and Lp(a) in the abstract. In fact, the journal's instructions authors limit to 200 words. Please see the Abstract completely revised.
Introduction
- COMMENT 1: The rationale for the study is not clear and a hypothesis is not stated.
- ANSWER 1: Thank you very much for this very relevant comment. We recognize that we have not translated our ideas well through clear and objective hypothesis. In fact, given the high number of parameters to be correlated in two different pathologies: metabolic (diabetes) and vascular (hypertension); we were much more focused on the nutritional aspect and so it was only the fatty acids ratio that guided our study.
- COMMENT 2: Please clarify the aim. Is it to look at associations or is it to predict presence of cardiovascular disease in the selected population? The different terminology used in title, aim and conclusion is confusing.
- ANSWER 2: Thank you for this important clarification about aim of the study. According previous comment, we completely rewrote the objectives and strengths conclusion points. This study is based on associations between the metabolic syndrome parameters and we proposed that the metabolic and vascular disorders could lead to thrombosis without real confirmation. Please see the MS completely revised.
- COMMENT 3: The first sentence is too complex and grammatically incorrect. Please revise. There are several other grammatical errors in the introduction.
- ANSWER 3: Thank you for this constructive comment. The introduction has been corrected mistakes grammatically and typographically. We have rewritten the first sentence and some parts of the introduction. Please see the MS completely revised.
- COMMENT 4: In line 39 the term food lipids should be replaced with dietary lipids.
- ANSWER 4: Thank you for the meaning and precision of the scientific terms. It's corrected. Please see the Track changes MS.
- COMMENT 5: Line 41 states various fats are correlated with a number of conditions. Correlated how? Do all have a positive correlation or a negative correlation?
- ANSWER 5: I agree. We sincerely appreciate this concern. We are very sorry for having not statistically analyzed carefully the correlation type. We redid the statistical analysis of all the data, taking account the correlation nature (positive or negative). We added the Table 4 which summarizes several correlation types between the fatty acids ratio (saturated and unsaturated) with the cardiometabolic syndrome parameters. Please see the table 4 and Track changes MS.
Methods
- COMMENT 1: How was the number of study participants decided? Was there a power calculation done?
- ANSWER 1: I agree. We are sorry for missing a detailed description of this information clearly in our methods section. Indeed we used a statistical formula to calculate the number of participants to recruit in each group. Please see the methods section in Track changes MS, but not detailed it.
The sample size is calculated using the following formula: N = e2 * p * (1- p) / i2
This formula was provided by the Algerian Ministry of Public Health (AMPH).
N: number of participants
e: 95% level will be 1.96
p: estimated proportion of the group population that exhibits the disease (T2DM and hypertension in our study) : 7% for T2DM (D group); 29% for Hypertension (H group); 40% for association T2DM and Hypertension (DH group). These percentages were provided by AMPH.
i: risk level fixed at 5% errors
Thus, we obtain the following results:
Group D: N= 1.96² × 0.07 × 0.93 / 0.05² = 100 T2DM participants
Group H: N = 1.96² × 0.29 × 0.71 / 0.05² = 316 Hypertensive participants
Group DH: N = 1.96² × 0.40 × 0.60 / 0.05² = 368 T2DM + Hypertensive participants
_________
100%: 0.93 + 0.07 for Group D; 0.29+0.71 for Group H; 0.40+0.60 for Group DH
- COMMENT 2: How was the dietary and nutrient info collected? Using 24 hour recall or FFQ? What foods were the participants consuming? What was the dietary total fat, PUFA, MUFA and SFA intake of all participants?
- ANSWER 2: I agree. Thank you for the constructive comments. As suggested by reviewer 1, we added the dietary questionnaire in methods section, data in results and discussed it. Also, we have presented the data of Table 4 (plasma fatty acid composition) to figures form. Please see the revised MS.
- COMMENT 3: What were the physical activity levels of the participants?
- ANSWER 3: I agree. Thank you for the interesting comment. We added the physical activity questionnaire in methods, data in results sections and discussed it. Please see the revised MS.
- COMMENT 4: Please report the brand of the insulin kit used.
- ANSWER 4: I agree. Thank you for this detail. We added the brand of insulin kit used. Please see the methods section revised of MS.
- COMMENT 5: Please replace the term diabetic patients to people with diabetes throughout.
- ANSWER 5: I agree. Thank you for this comment. We made this change in all MS and even in the title of this submission.
- COMMENT 6: Line 144 'hypertension with diabetic' is grammatically incorrect and should be replaced with hypertension with or without type-2 diabetes.
- ANSWER 6: I agree. Thank you for precision of this comment. We made this change in all MS. Please see the revised MS.
Results
- COMMENT 1: Throughout the results the term increased or decreased should be replaced with higher or lower. When comparing different groups of people the term increase would be incorrect. And increase or decrease is measured from a baseline value not between groups.
- ANSWER 1: I agree. Thank you for the meaning and precision of the scientific terms. It's corrected. Please see the Track changes MS.
- COMMENT 2: In Table 1 please give a total number of participants (N) in each group.
18. ANSWER 2: I agree. Following the response of comment 11 (methods section), we added a total number of participants (N) in each group. Please see the Table 1 revised.
- COMMENT 3: Why are the participant characteristics been reported separately for males and females? The aim of the study was not to look at sex differences and not all results have been reported separately for male and females. All results should be reported combined for both genders unless the aim is to look at sex differences.
- ANSWER 3: I’m totally agreed with the reviewer. However, according to the NCEP-ATPIII definition (our study), we will maintain the distinction between male and female only for 02 parameters: waist circumference and HDL-cholesterol. All other parameters are common between both sexes. Moreover, we randomized females and males together and not separately. The data has been corrected in Table 1. Please see the Track changes MS.
- COMMENT 4: The term Man and Women should be replaced with males and females in table 1.
- ANSWER 4: I agree. I’m sorry for this confusion scientific language term. It's corrected. Please see the Table 1 and also in Track changes MS.
- COMMENT 5: A lot of the interpretation of results is based on correlation analysis.
I would suggest to make a table for all the correlations that were calculated in the analysis and provide p values in that table.
- ANSWER 5: I agree. Following this suggestion, we added a table for most relevant correlations. Please see the Table 4 in Track changes MS.
- COMMENT 6: Also a lot of comparison is done between D, H and DH groups, however the result tables only show statistics ‘compared to control group’. The authors could use a superscript letter method of showing the statistical differences between various groups?
- ANSWER 6: I agree. Thank you for the interesting comment. As suggested, we have added in the methods section (statistical analysis) and in the results tables, the superscript letters to compare the 03 groups between them: Control (a), D (b), H (c) and DH (d). Please see the table 1, 2 and 4 in Track changes MS.
- COMMENT 7: Line 59 and 60. It is not clear what the terms dominant adipose tissue and female predominance mean?
- ANSWER 7: I agree. We apologize do not being clear in this observation. In fact, in this study, both in women or men, the fat mass is localized mainly in the abdominal or visceral region. While in healthy subjects, males do not exhibit abdominal expansion and females show fatty deposits in the pelvic region, but not the abdominal region. Please see the Track changes MS.
- COMMENT 8: % of body fatness should be replaced by body fat percentage (% BF).
- ANSWER 8: I agree. Thank you for this comment. We made this change in table 1 and in all MS
- COMMENT 9: Line 65 and 91 an r value has been reported but not a p value? Please report the p values.
- ANSWER 9: I agree. Thank you for this statistical analysis precision. We reported the p-value in all revised MS
- COMMENT 10: Line 68 increased insulin secretion should be replaced by fasting circulating insulin levels. The authors did not measure secretion.
- ANSWER 10: I agree. I’m sorry for this scientific language confusion. It's corrected. Please see the Track changes MS.
- COMMENT 11: Table 4 should specify ‘plasma’ in the title.
- ANSWER 11: I agree. I'm sorry to do not specify that the assay was done in plasma. Please see the figures that replaced table 4 (as suggested by reviewer 1).
Discussion
- COMMENT: Again there are many errors in the use of English language in the discussion.
28. ANSWER: I agree. The revised version of the MS has been checked duly and thoroughly for the grammatical and typographical mistakes. We have revised our MS by a native English speaker (Abstract + MS). Please see the revised MS. However, if the Editor-in-Chief decides to correct the MS by the MDPI editing English correction system (Regular grammar check), we agree to pay the correction fee.
Conclusion
- COMMENT 1: I am not convinced by the conclusion
29. ANSWER 1: I agree. After adding other data (dietary lipids screening, physical activity, pro inflammatory cytokine) and discuss them, the conclusion was completely reflected and rewritten. I hope she will be convinced.
- COMMENT 2: While I agree the study shows an association between the FA ratios in the plasma and some of the metabolic complications, the authors cannot prove the unbalanced ratio occurred before the metabolic syndrome symptoms. So it’s not possible to prove that the plasma FA ratios are ‘predictive’. There is just an association.
- ANSWER 2: Thank you for this very relevant comment. As suggested, we have maintained that only the keyword "association" and not predictive. However we hypothesized that the predictive atherogenic and thrombogenic effect by fatty acids ratio unbalance, has proposed in conclusion. It is really that we highlighting it in our daily consultations saw the changed lifestyle in people.
Please see the Track changes MS.

Round 2
Reviewer 2 Report
The authors have made significant improvements to the manuscript. I have some minor concerns
Abstract line number 33 “is associated to lipid” should be changed to “associated with lipid”.
Abstract line 55 Conclusion “the unbalance ratio” should be changed to “the unbalanced ratio”. The whole sentence does not make sense and needs correction. “seems predisposed to thrombosis in diabetics people” means what? Are the authors suggesting the unbalanced ratio may lead to increased risk of thrombosis? Please simplify the sentence to make it clearer. Also change the word diabetic’s people to people with diabetes.
Introduction Line 75 Please change the word non-glucodependent to non-glucose dependent.
Methods: It seems two different methods were used to obtain dietary data: FFQ and 24 hour recall. Please explain how the data was analysed from these two different methods? Was it combine and averages were taken of different methods were to calculate different nutrients?
Methods line 328 states The discontinued medication was ‘fixed’. What does that mean?
Results I am glad the authors have agreed to show differences between groups, but it is not clear what different superscripts are reporting? When a group has all 3 superscripts a,b,c what does that mean? Please explain in detail in table legends.
Results The graphs seems to use ** but the figure legends do not explain what the ** stand for.
Conclusion. The whole paragraph on conclusion is very confusing. And it’s because of incorrect grammar. It makes it difficult to interpret. Please correct all sentences of the conclusion paragraph.
